# Wild salmon migration routes influence sea lice infestations: An agent-based model predicting farm-related infestations on juvenile salmon

**Jaewoon Jeong**[1]*, **Gregor McEwan**[2]

1 Aquaculture, Biotechnology and Aquatic Animal Health Science Branch, Fisheries and Oceans Canada, Ottawa, ON, Canada, 2 Modail Mara Inc., Charlottetown, Prince Edward Island, Canada

* jwjeong79@gmail.com

**Data Availability Statement:** https://github.com/jaewoonjeong/ABMsalmon.

**Funding:** Aquaculture, Biotechnology and Aquatic Animal Health Science Branch at the Fisheries and

## Abstract

This study presents an Agent-Based Model (ABM) simulation to assess the impact of varying migration routes on sea lice (*Caligus clemensi*) infestation levels in juvenile wild sockeye salmon (*Oncorhynchus nerka*) in the Discovery Islands, British Columbia, Canada. This research highlights the importance of migratory routes in determining the extent of exposure to sea lice originating from nearby salmon farms. Three northward out-migration routes were modelled, each exposing the fish to different levels of infestation pressure based on proximity to salmon farms. The ABM incorporates spatially explicit migration patterns of juvenile sockeye salmon using a detailed raster map of the Discovery Islands. Key variables such as swimming speed, progression rate, and infestation levels were integrated into the model, offering a comprehensive analysis of migration and infestation dynamics. The study revealed that infestation rate is highly variable, depending on migration routes. Specifically, salmon traveling longer migration routes with lower infestation pressure may experience higher sea lice loads compared to those on shorter routes with higher infestation pressure. This underscores the role of low infestation pressures and the critical influence of swimming speed, which affects exposure time, and thus infestation rates. Additionally, the study conducted a sensitivity analysis to understand the influence of various parameters on infestation rates. This analysis highlighted the importance of swimming speed and progression rate, particularly in routes closer to the farms. The findings suggest that slower swimming speeds and meandering routes increase exposure to lice, thereby elevating infestation levels. The research contributes to understanding the dynamics of sea lice transmission and its relationship with salmon migration patterns. It underscores the necessity of considering migratory routes and farm proximity in managing and mitigating the impact of sea lice infestation on wild salmon populations. This study's insights are crucial for developing strategies to balance aquaculture practices with the conservation of wild salmon.

Oceans Canada provided a research supporting fund for this study. The funders had no role in study design, data collection and analysis, decision to publish, or preparation of the manuscript.

**Competing interests:** The authors have declared that no competing interests exist.

## 1. Introduction

Maintaining aquaculture sources like salmon farming is crucial to meet the growing seafood demand [1]. However, salmon aquaculture has been linked to the declining populations of wild salmon [2]. A major concern is the transmission of sea lice from salmon farms to migrating juvenile wild salmon; juveniles are more vulnerable than adults due to their underdeveloped scales and immune systems, which offer less protection against parasites [3]. Such infestations in juvenile salmon may lead to a diminished number of adult salmon returning to their spawning grounds. Consequently, it becomes imperative to thoroughly evaluate the risk that salmon farms pose to wild salmon populations and implement measures to mitigate this risk while meeting the rising demand for salmon as a food source.

It is challenging to predict sea louse infestation in juvenile salmon during their migration through areas with sea lice presence. This complexity arises from an interplay of factors such as sea lice population dynamics, the speed and distance of salmon migration, and the effectiveness of lice attachment to fish [4]. The complex behaviour of juvenile salmon during migration further complicates experimental trials. Nevertheless, computer models, especially spatially explicit Agent-Based Models (ABM), offer a viable solution for simulating intricate spatial scenarios, such as salmon migration [4]. ABMs are valuable for incorporating the spatial aspects of the system and for simulating the stochastic movement of wildlife [5]. They are considered a promising tool for representing the stochastic nature of fish movement [6] and have been used to simulate the stochastic characteristics of wild juvenile salmon migration [7]. For instance, ABM can incorporate individual variability in progression rate and route choice [8]. An ABM was used to explore how environmental and intrinsic factors can modulate the effect of salmon lice on survival, growth and maturation of migrating Atlantic salmon smolts [9], as well as on the population dynamics of sea trout [10].

Infestation pressure is a crucial indicator of the lice infestation threat from farms to wild juvenile salmon. It is calculated by estimating the number of infective lice larvae in the water column, often employing data from lice counts on farmed salmon and models that simulate larval dispersion, based on environmental elements such as currents and tides [11]. However, the accurate quantification of sea lice infestation pressure presents challenges due to the complex life cycle of sea lice, their uneven spatial distribution of larvae and environmental variability [12]. Despite these challenges, the integration of direct measurement with modelling has been enhancing the precision of these estimates [13]. Models estimate sea lice infestation rates at a particular location by calculating the infestation pressure, considering the number of sea lice at neighbouring farms and their proximity to these farms [9, 14, 15]. In determining the relationship between infestation pressure and the infestation rate on juvenile salmon, lice counts from sentinel cages or trawls were used to align with the assessed infestation pressure [16].

In this study we developed an ABM to simulate the out-migration of juvenile sockeye salmon in the Discovery Islands, British Columbia (BC), Canada. Sockeye salmon (*Oncorhynchus nerka*), after one or two years in freshwater, mature into smolts adapted for saltwater life and migrate downstream towards the ocean [17]. This species is the dominant Pacific salmon passing through the Discovery Islands region in BC [18]. Recent research utilized acoustic tagging to gain deeper insights into these migration patterns [19]. Their findings indicate significant variability in the distance travelled and the speed of migration, depending largely on the chosen routes. The unique geographical features of the Discovery Islands, with their channels and peninsulas, offer a variety of migration seaways. The choice of these routes can lead to varying levels of exposure to nearby salmon farms, suggesting that the fine-scale migratory routes are crucial in determining the rate of lice infestation [20].

In this study, we present results from an ABM simulating juvenile sockeye salmon migration through multiple fine-scale routes in the Discovery Islands, BC, Canada. These routes expose the salmon to larval lice originating from nearby salmon farms. Our model evaluates how different migration routes affect sea lice infestation rates on out-migrating wild juvenile salmon. Additionally, we conduct sensitivity analysis to identify key factors influencing the infestation rate. The analysis helps us understand the uncertainty and potential implications of various parameters on the rate of infestation, thereby offering a deeper insight into the dynamics of sea lice transmission in this context.

## 2. Materials and methods

The model, developed in R [21], is an agent-based simulation that captures the out-migration of juvenile wild salmon in an offshore region, including potential infestations by larval lice during their migration. The source code is available at https://github.com/jaewoonjeong/ABMsalmon.

To describe our model, we have used the ODD (Overview, Design concepts, and Details) protocol [22]. In Overview, we introduce the model's agents. In Design concepts, we describe the general principles of the model's design. Details explains how fish migrate in the region and how they are infested with lice, including the equations that govern how the model operates.

### 2.1. Overview

**2.1.1. Purpose.**   The primary objective of our model is to examine the impact of different out-migration routes on sea lice infestation levels in juvenile wild salmon. Focusing on a case study, we consider the scenario in which juvenile sockeye salmon encounter *Caligus clemensi* while migrating through the Discovery Islands, BC, Canada. We specifically consider *C. clemensi* as the primary infesting species over *Lepeophtheirus salmonis*, as it is the dominant lice species found on sockeye salmon in this region [18, 23]. Although our model is tailored to specific louse and salmon species within certain regions, it can be broadly applicable to various settings, aiming to understand the impact of salmon migration on lice infestation rates. The model highlights how varying migration routes lead to differing degrees of exposure to sea lice from nearby salmon farms. Thus, we simulate the salmon navigating through these routes to investigate the consequent impact on infestation levels by the end of their migration. Additionally, through a sensitivity analysis, we explore the influence of various migration-related parameters and proximity to salmon farms on the abundance of lice in wild salmon.

**2.1.2. Entities, state variables and scales.**   The entities of the models are individual juvenile sockeye salmon. The state variables describing the salmon include fish location (longitude and latitude), swimming speed (km/day), progression rate (km/day), migration duration (days), calendar date since the start (days since June 1st), migration distance (km), and infestation level (number of lice per fish).

Our model provides a spatially explicit depiction of the offshore migration routes of juvenile sockeye salmon in the Discovery Islands (**Fig 1**). To achieve this, we created a detailed raster map of the Discovery Islands, with each pixel measuring 100m x 100m, enabling comprehensive quantitative mapping along the coastline [14]. This mapping utilized the R package 'raster' [24]. Each 10000 m$^2$ pixel is characterised by infestation pressure, which indicate the concentration of *C. clemensi* copepodid larvae present. The mechanism by which salmon entities increase their infestation level upon encountering these larvae is detailed below.

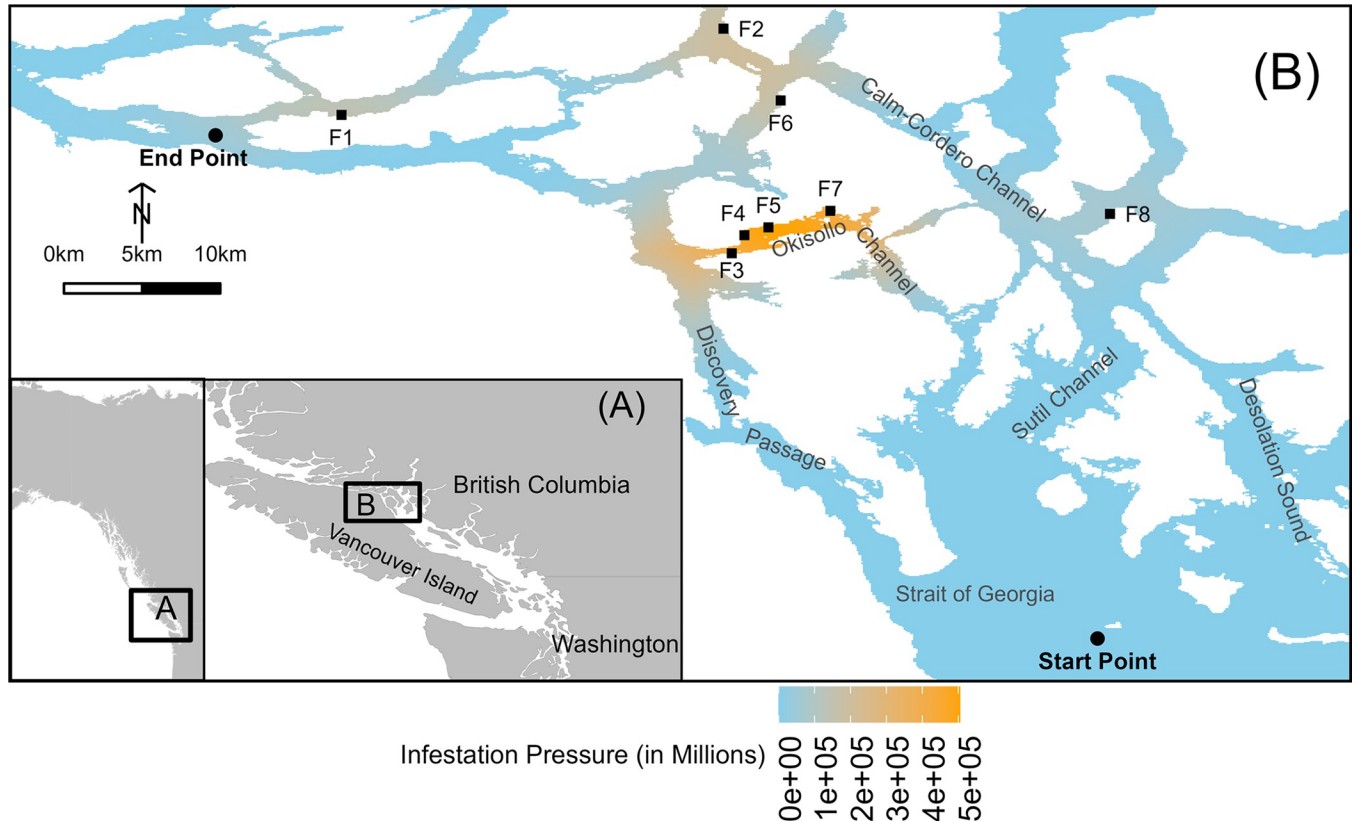

**Fig 1. The study region on the Discovery Islands, British Columbia, Canada.** The map displays eight salmon farms (F1-F8) represented as black squares. Juvenile sockeye salmon are simulated to migrate from the Start Point to the End Point. Infestation pressure refers to the local concentration of *Caligus clemensi* copepodid larvae per 100m x 100m grid cell. The maps in the insets were created using R with public domain Natural Earth base maps (http://www. naturalearthdata.com/), and the main map was created using R with public domain British Columbia Data Catalogue base maps (https://catalogue.data.gov.bc. ca/).

Typically, sockeye salmon from the Fraser River embark on their journey into the Strait of Georgia during spring [23]. Once in the ocean, their migration pattern generally leads them northward towards the North Pacific Ocean. They usually reach the Discovery Islands area around May and June [25]. We based our design of the three northward out-migration routes on the findings of research that tracked acoustically tagged juvenile sockeye salmon as they navigated through the Discovery Islands [19]. In our simulations, all individuals are assumed to be released on the same day: June1ˢᵗ. This assumption simplifies the out-migration of juvenile sockeye salmon in Discovery Islands originating from Fraser River. However, since the infestation pressure in our model does not include temporal dynamics, this simplification has minimal impact on the final outcome.

**2.1.3. Process overview and scheduling.** We established three out-migration routes, each allocated 3000 fish, between the start and end points. To estimate the salmon's swimming speed, we assumed that fish traveling through Discovery Passage follow the most direct seaway distance without meandering. Based on this assumption, we calculated a baseline swimming speed of 31.1 km/day. This calculation is derived from the average speed of sockeye salmon traversing the Discovery Passage in 2017 and 2018 [19]. This speed is consistent with the previously observed range of 0.2–0.5 m/s [26].

Given our model's specific focus on examining the impact of different migration routes, we intentionally omitted certain biological characteristics of juvenile wild fish, like body size and

body growth. This deliberate exclusion is based on the rationale that such factors would uniformly affect fish across all migration routes, and our primary interest lies in isolating and analysing route-specific impacts.

## 2.2. Design concepts

This section outlines the fundamental principles underlying the model's design. Within the framework of the ODD protocol, we employ to describe our model, various topics are defined in the design concepts section, including Emergence, Adaptation, Objectives, Sensing and Stochasticity.

**2.2.1. Emergence.** We are interested in one emergent property of the model: the infestation of sea louse on juvenile wild salmon during their out-migration.

**2.2.2. Adaptation.** Individual salmon are allowed to move to one of four pixels of north, south, west or east.

**2.2.3. Objectives.** Individuals are simulated to migrate towards the endpoint using one of three possible out-migration routes. The specific routes taken by individual fish may differ due to stochastically modelled movements.

**2.2.4. Sensing.** Individual fish are assumed to be able to sense the seaway distance from those four possible next locations to the end point. They use these distances to make the probability-weighted sampling, which determines their next location of pixel.

**2.2.5. Interaction.** Our model only investigates the impact of different migration routes, so the salmon do not interact with each other. Any influence that other salmon may have on each other's paths are simplified to be included in our 'meandering' variable, which gives us explicit control.

**2.2.6. Stochasticity.** Empirical observational studies using tagging approaches have shown great variability in migration speed. Therefore, we added a stochastic element to the model to create individual variability in migration speed and fine-scale migration seaways [27], which we refer to as 'meandering'. The mechanism works as follows: based on the distance from four points to the endpoint, the salmon are most likely to choose the location with the shortest distance to the endpoint. The location with the second shortest distance has the second highest probability, and so on. We varied the distribution of probabilities for these four potential next locations on each route to align the model's migration distance and period with the actual observational data of sockeye salmon in the Discovery Islands [19]. An advantage of making meandering an adjustable input variable is that it enables us to explore the effects of varying spatial aspects of salmon swimming behaviours on infestation levels.

Moreover, the sea lice infestation in our model is treated stochastically, being determined by a negative binomial distribution. The infestation pressure in each pixel serves as the mean parameter for this distribution, a process which is elaborated in Section 3 'Details'.

**2.2.7. Observations.** The primary output from the model is the count of sea lice infestations on individual salmon. In addition, the model records the period and distance of their migration.

## 2.3. Details

**2.3.1. Initialization.** In each simulation run, an individual fish begins its migration at the starting point and swims at a speed of 31.1 km/day. The fish is deterministically assigned one of the three routes.

**2.3.2. Input data.** The model's input includes infestation pressure, imposed by all eight salmon farms, applied to each pixel within the study region.

**2.3.3. Sub-models.**  *2.3.3.1. Migration model.* During May and June of 2017 and 2018, eight active Atlantic salmon farms were identified in the study region [28], coinciding with the period when tracked the migration of juvenile sockeye salmon in the Discovery Islands [19] (**Fig 1B**). Of these, four farms were clustered in the Okisollo Channels, while three were more dispersed in the Calm-Cordero Channel. Notably, no farms were present in Discovery Passage. One farm is positioned near the endpoint. As a result, each of the three migration routes exposed the salmon to varying degrees of proximity to these farms. Route 1 offered the least exposure, being relatively distant from the farms. In contrast, Route 2 passed very close to four farms, and Route 3 mostly avoided immediate proximity to the farms.

In the model, we assumed that traversing each pixel represents a uniform 100-meter journey, either in a north-south or east-west direction, explicitly excluding diagonal paths (**Fig 2**). This design is based on the 10000 square meter size of each pixel [14]. To navigate, individual fish use a raster map formatted into a 100m x 100m grid. In the open ocean segment of the model, each pixel is assigned a seaway distance value to the endpoint [27], which the fish use to stochastically determine their migration route. At each movement step, the fish assess distances to the endpoint from their four adjacent cells to select their subsequent direction.

If fish were to take the shortest seaway path from start to end point, the migration distances for Routes 1, 2, and 3 would be 106.7 km, 113.7 km, and 143.7 km, respectively, corresponding to migration durations of 3.4 days, 3.7 days, and 4.6 days. However, realistically, fish hardly follow such direct routes and often exhibit meandering behaviour, introducing randomness to their movement. Consequently, our model assumed that fish possessed an innate ability to navigate efficiently towards their feeding grounds, while also allowing for deviations from

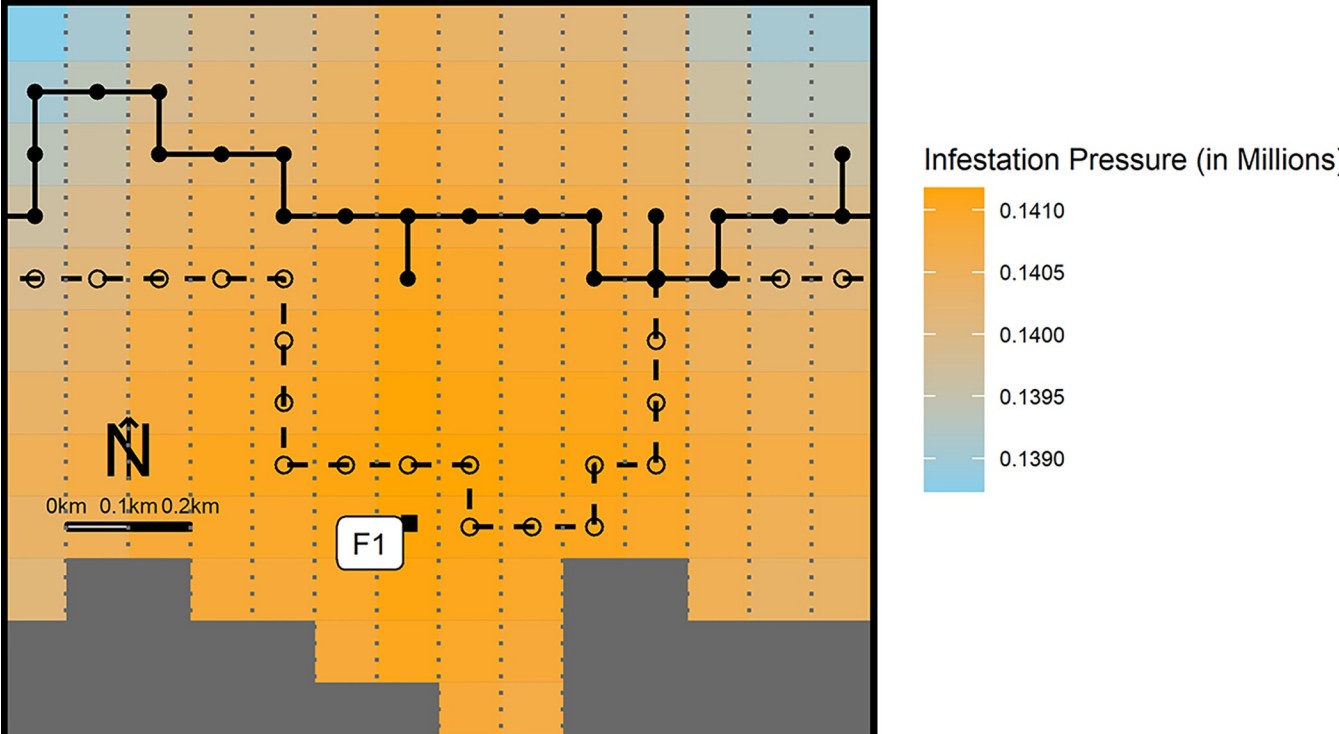

**Fig 2. Two examples of stochastic migration paths of juvenile sockeye salmon.** These paths demonstrate their journeys along Route 3, near farm F1 (as detailed in Fig 1). The sample salmon travelled from east to west, with closed and open circles indicating the pixels they traversed. Each pixel in the grid measures 100m x 100m. The salmon spent approximately 4.63 minutes in each cell, maintaining an equal duration across all cells.

their path to account for random movements [20]. This approach aimed to balance the inherent randomness of natural fish movements with the purposeful behaviour believed to benefit them in reaching specific geographic destinations.

To account for this variability in movement, we integrated concepts of swimming speed and progression rate into our model. The swimming speed was defined as the distance travelled between two points divided by the migration period [8], expressed as:

$$Swimming\ speed = \frac{D_{lm}}{t} \tag{1}$$

The progression rate was calculated as the shortest seaway distance between two points divided by the migration period, represented as:

$$Progression\ rate = \frac{d_{lm}}{t} \tag{2}$$

Here, $D_{lm}$ denotes the actual movement distance of fish between two points of $l$ and $m$, $d_{lm}$ signifies the shortest seaway distance between these two points, and the migration period, $t$, is consistently applied to both calculations.

When a fish takes the most direct seaway route between two points, its swimming speed matches the progression rate. However, if the fish opts for a route that deviates from the most direct seaway path, the distance covered, as indicated by its swimming speed, will be greater than the progression rate. As such, swimming speed can be considered a theoretical and perhaps less pragmatic measure when compared to the progression rate. A significant disparity between the swimming speed and progression rate indicates prolonged meandering by the fish during its migration, resulting in covering extra distance. Given that the swimming speed remains consistent throughout each simulation, the migration distance is directly proportional to the migration period in our baseline model.

Assumptions about differing progression rates across migration routes were informed by observations from Rechisky *et al.*, which tracked acoustically-tagged juvenile sockeye salmon during the out-migration periods of 2017 and 2018 [19]. In these respective years, 364 and 150 salmon were released, with 89 and 116 subsequently detected in the Discovery Islands. The deployment of multiple receiver arrays along probable migration routes in the region enabled a detailed analysis of the migration patterns of juvenile sockeye salmon by multiple routes in the area. The observed fish exhibited average progression rates of 31.1, 18.1 and 12.1 km/day for routes 1, 2, and 3, respectively [19]. Since only progression rates were recorded for acoustically-tagged juvenile sockeye salmon, we assumed a uniform swimming speed of 31.1 km/day, in line with the observed progression rate for Route 1 [19]. In our model, we designate one time step as 4.63 minutes, during which a fish traverses 0.1 km. The swimming speed is consistent across all routes throughout the simulation. We introduced variation in progression rates using probability-weighted sampling, influenced by the meandering parameter, α. As fish decide which next pixel to move to among four possible directions (north, south, west, and east), the choice probability is dictated by the distance to the endpoint, with the pixel at the shortest distance being the most likely selection. For the upcoming four pixels in the subsequent time step, the probability for each is defined in the sequence of 1-α, α/2, α/2, and 0, based on their increasing distances to the endpoint. The α values were set at 0, 0.29, and 0.40 for Routes 1, 2, and 3, respectively, calibrated to match the model's average progression rates with those observed [19].

*2.3.3.2. Infestation model.* The infestation model works with the migration model to determine the likelihood of infestation of the migrating wild juvenile sockeye salmon. In a simplified approach, the infestation likelihood is determined by multiplying the attachment rate,

which indicates the frequency of successful lice infestation post-contact, with the encounter rate, representing the frequency of contact between fish and lice [29]. The attachment rate is influenced by sea lice biology, host fish attributes, and environmental conditions [30]. In contrast, the encounter rate hinges on the sea lice concentration. To emphasize the impact of salmon farms, we excluded variables associated with the attachment rate, assuming every encounter results in infestation.

Price *et al.* found significantly higher *C. clemensi* abundance in salmon migrating through salmon farms in the Discovery Islands compared to those in a salmon farm-free area on the north coast of BC, Canada [23]. Furthermore, they observed a marked increase in *C. clemensi* abundance in salmon after passing through the farms in the Discovery Islands, suggesting that open net-pen salmon farms may facilitate the transmission of *C. clemensi* to wild juvenile sockeye salmon. Therefore, in our model, we assumed that all lice infestations originated from *C. clemensi* copepodids associated with salmon farms to specifically examine the impact of salmon farms on lice infestation in juvenile sockeye salmon.

The output of our ABM is the sea lice count on fish at the migration endpoint. This count is derived as the cumulative sum of random numbers assigned at each pixel that the fish traversed. These numbers are stochastically selected from a negative binomial distribution, which is characterized by its mean ($\mu$) and a dispersion parameter ($\kappa$). Consequently, the total sea lice count results from the summation of these numbers throughout the migration process. We used $\mu_i$ to represent the number of lice encounters at pixel i, and $\kappa$ was set at 0.921 [15]. Infestation pressure represents the number of infective larvae at a specific pixel [15]. $\mu_i$ was derived from the function of infestation pressure, $IP_i$, at pixel $i$, in relation to swimming speed. The calculation of $\mu_i$ followed this equation [15].

$$\mu_i = \exp\left(-5.49 + log\left(\frac{0.1}{Speed}\right) + 0.425*log(IP_i)\right) \tag{3}$$

Here, 'Speed' represents the swimming speed of juvenile sockeye salmon at 31.1 km/day. The term $log(0.1/Speed)$ adjusts the equation to account for the duration a migrating fish spends at a pixel. $IP_i$, which represents the infestation pressure at pixel $i$, offers an estimate of the copepodids present with a fish for a specific duration at that pixel. Eq 3 was derived from regression models by Stige *et al.*, examining salmon lice (*L. salmonis*) abundance as a function of infestation pressure on juvenile Atlantic salmon (*Salmo salar*), based on data from post-smolt trawling and sentinel cage experiments [15]. This logarithmic relationship between infestation pressure and lice abundance, identified as the best-fitting model, is henceforth referred to as the 'logarithmic model' [15]. In contrast, an alternative approach proposed a proportional change in lice abundance corresponding to changes in infestation pressure and used a fixed slope between these variables, as shown in Eq 4:

$$\mu_i = \exp\left(-11.6 + log\left(\frac{0.1}{Speed}\right) + log(IP_i)\right) \tag{4}$$

This equation was also employed to simulate our model, exploring the significance of the presumed relationship between infestation pressure and lice abundance on our modelling outcome. We refer to this approach as the 'proportional model'.

The cumulative infestation pressure across all traversed pixels provides a comprehensive measure of the infestation risk to the fish during its out-migration, originating from copepodids associated with the eight salmon farms in the region. To quantify infestation pressure, specifically the number of copepodids at a distinct pixel, we utilized the total number of copepodids from all the farms, applying a kernel density function [31]. Infestation pressure $IP_i$ at

pixel $i$, resulting from all the farms in the study region, was calculated using the following equation,

$$IP_i = \sum_{j=1}^{8} X_{i,j} \tag{5}$$

The pressure exerted from farm $j$ at pixel $i$ ($X_{i,j}$) was derived from,

$$X_{i,j} = \frac{W(d_{i,j})}{\sum_{i=1}^{n} W(d_{i,j})} * N \tag{6}$$

Here, "n" denotes the total pixel count. "N" refers to the number of copepodid lice per farm. The copepodid tally from the farms gets allocated across pixels, contingent upon the seaway distance between each farm and all the study area locations, utilizing a kernel density function [4]. This equation employs the Gaussian kernel density to apportion copepodid numbers from farms based on the seaway distance from farm j to pixel i.

In our model, all farms maintain a uniform number of sea lice, which remains constant throughout the simulations. We assumed that each farm releases an identical total of 62.5 million copepodids, summing to 500 million from the eight farms. This value was selected to ensure that our model replicates sea lice abundance levels comparable to those in Price *et al.* [23].

The function $W(d_{i,j})$ was derived using the Gaussian kernel density function. This function reflects the diminishing weight as the seaway distance between locations increases.

$$W\left(d_{i,j}\right) = \frac{1}{\sqrt{2\pi}(\sigma/4)} e^{-\frac{d_{i,j}^2}{2(\sigma/4)^2}} \tag{7}$$

Here, $W$ represents the Gaussian kernel density's estimated weight for the seaway distance $d_{i,j}$, which signifies the seaway distance between the pixel $i$ and salmon farm $j$. We assumed a fixed bandwidth ($\sigma$) of 30 km, representing the maximum distance of larval lice dispersal from their origin point [32]. Seaway distances between each combination of pixels and farms were calculated using the 'gdistance' package [33] in the R statistical language.

*2.3.3.3. Sensitivity analysis.* We performed a sensitivity analysis on our model to assess the impact of key parameters on the outcomes, specifically the counts of sea lice on fish at the endpoint. The parameters analysed included the chosen migration route and four additional key parameters: i) number of sea lice released from farms, ii) bandwidth within the Gaussian kernel density function, iii) swimming speed, and iv) progression rates. The number of sea lice and the bandwidth were selected for the analysis due to their critical roles in estimating infestation pressure, while swimming speed and progression rate were considered because they markedly affect the duration of salmon exposure to sea lice. For our sensitivity analysis, the values for each parameter were varied using a uniform distribution within their specified ranges.

We summarized the observations of *C. clemensi* abundance on the farms in the Discovery Islands during the out-migration period of wild juvenile Pacific salmon from 2011 to 2022 [28]. To encompass a broad spectrum of probable fluctuations in copepodid numbers from farms, we calculated the proportions of the 1st and 3rd quartiles to the median value, which were 0.18 and 4.12, respectively. These proportions were then applied to the baseline number of sea lice released from farms to determine the range of lice numbers used in our sensitivity analysis. A reasonable range of bandwidths from 5 km to 100 km was explored [32]. Based on observed variations in travel speeds among individual sockeye salmon during coastal

migration in southern BC, Canada [8, 34], the swimming speeds in our simulations varied from 6.2 km/day (0.2 times the baseline speed) to 46.7 km/day (1.5 times the baseline speed) [35].

Unlike the number of sea lice, bandwidth, and swimming speed, the range of progression rates was not predetermined because these rates were established after simulating the salmon migration in our model. Through 3000 simulations of the baseline model, we identified the minimum and maximum progression rates, which were subsequently used to define the range for our sensitivity analysis. Then, we determined which values of the meandering parameter ($\alpha$) corresponded to specific progression rates. This approach allowed us to explore varying progression rates in the sensitivity analysis by adjusting the values of $\alpha$. Furthermore, varying progression rates were applied only to Routes 2 and 3, as it was assumed that fish followed a direct path without randomness in Route 1.

In our sensitivity analysis, simulations were conducted by jointly and randomly selecting values for each of the four key parameters within their pre-defined ranges, resulting in a total of 3000 simulations where all parameters varied randomly. To evaluate the sensitivity of the model's outcomes to different parameters, we utilized Pearson's correlation coefficient ($r$). This coefficient measures the linear relationship between a model parameter and its output, with values spanning from -1 to 1. A value of 1 indicates a direct proportional increase, $-1$ reflects an inverse relationship, and a value near 0 suggests a negligible or weak correlation. Subsequent to the initial sensitivity analysis, which covered all routes and four key parameters, we conducted an in-depth investigation into the sensitivity specific to each route. This was to gain a clearer understanding of how the characteristics of each route uniquely impact the overall model.

## 3. Results

### 3.1. Juvenile salmon out-migration

**Fig 3** presents ABM simulation depicting out-migration examples along three routes for juvenile sockeye salmon in the Discovery Islands. Unlike Route 1, where all simulated fish deterministically took the shortest seaway of 106.7 km, the migration distances for Route 2 ranged from 153.4 to 282.7 km and for Route 3 ranged from 281.5 to 513.7 km (**Fig 4A**). The mean migration distances for Routes 2 and 3 were 196.8 and 373.3, respectively (**Table 1**). Thus, fish through Route 2 and 3 in our model took on average 1.73 (196.8 km / 113.7 km) and 2.60 (373.3 km / 143.7 km) times more migration distance than the shortest seaway of each route, respectively. While the progression rate for Route 1 was 31.1 km/day in every simulation, the

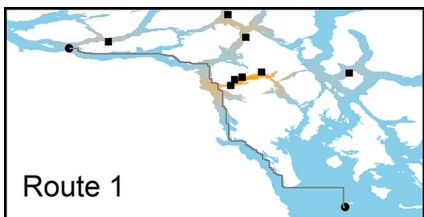
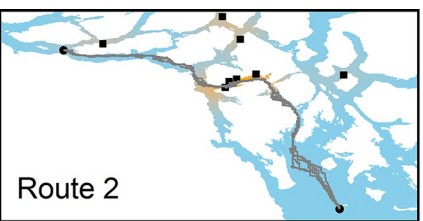
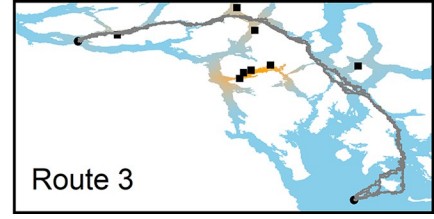

**Fig 3. Example of five juvenile sockeye salmon stochastic migration paths for each route in the Discovery Islands, British Columbia, Canada.** The map marks eight salmon farms with black squares and delineates each migration route from the start (bottom-right black circle) to the end (upper-left black circle). In every simulation, Route 1 deterministically adheres to the shortest path, resulting in an identical route marked by straight lines and right angles across all simulations. Conversely, Routes 2 and 3 show meandering salmon migration patterns, attributable to the stochastic nature of their path selection. Infestation pressure is applied in a manner identical to that shown in Fig 1. These maps were created using R with public domain British Columbia Data Catalogue base maps (https://catalogue.data.gov.bc.ca/).

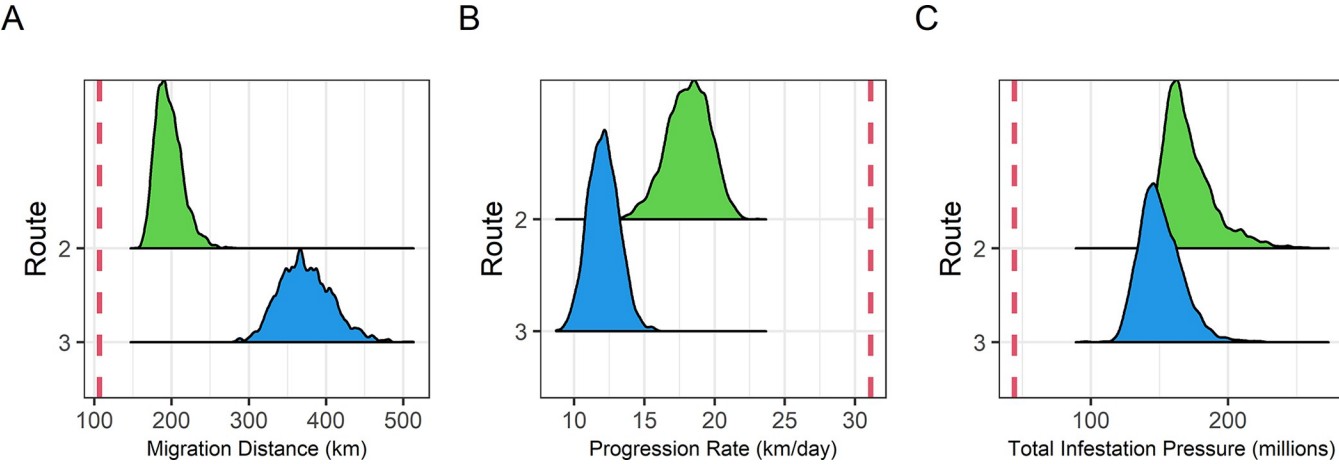

**Fig 4. Migration metrics distribution across simulations.** For 3000 simulations of an agent-based model, the distributions of (A) migration distance, (B) progression rate, and (C) total infestation pressure are presented. In all simulations, Route 1 showed consistent results, marked by vertical dashed red lines.

progression rates for Route 2 ranged from 12.5 to 23.1 km/day and for Route 3 ranged from 8.7 to 15.9 km/day (**Fig 4B**).

Total infestation pressure, defined as the cumulative number of lice encountered by salmon across each pixel along their path, of Route 1 was 44 million, and total infestation pressure of Route 2 and 3 ranged from 136 million to 274 million and from 95 million to 224 million, respectively (**Fig 4C**). Although Route 2 has salmon farms in closer proximity than Route 3, its total infestation pressure was only marginally higher. This is attributed to Route 3's longer migration distance, resulting from both its inherently longer seaway and its slower progression rate. **Fig 5** illustrates the cumulative infestation pressure per pixel for out-migrating fish from start to end, based on five examples of the simulations. Fish on Route 1 faced a sharp increase in infestation pressure near the four farms in the Okisollo Channel (**Fig 1B**), but their exposure duration was the shortest and subsequently the cumulative infestation pressure was smaller than the other two routes. Fish on Route 2 encountered the most substantial rise in infestation pressure, especially when they migrated through the four farms in the Okisollo Channel. Conversely, those on Route 3 had a notable increase in infestation pressure as they moved through the Calm-Cordero Channel, but this was not as a sharp increase as Route 2. Yet, fish on Route 3 had a longer exposure, covering a distance about twice that of Route 2. Notably, the variations between routes were much more noticeable than the individual variations within the same route.

## 3.2. Sea lice abundance by migration routes

In general, fish on Route 1 exhibited markedly lower abundance, and fish on Route 2 yielded slightly lower sea lice abundance than fish on Route 3 (**Fig 6**). In the logarithmic relationship,

**Table 1. Mean migration distance, migration duration, swimming speed, progression rate and total infestation pressure for each route.** These values were derived from the averages of 3000 simulations per route.

| Route | Migration distance (km) | Migration period (days) | Swimming speed (km/day) | Progression rate (km/day) | Total infestation pressure (in millions) |
|-------|------------------------|------------------------|------------------------|--------------------------|------------------------------------------|
| 1 | 106.7 | 3.4 | 31.1 | 31.1 | 44.3 |
| 2 | 196.8 | 6.3 | 31.1 | 18.1 | 171.1 |
| 3 | 373.3 | 12.0 | 31.1 | 12.1 | 150.8 |

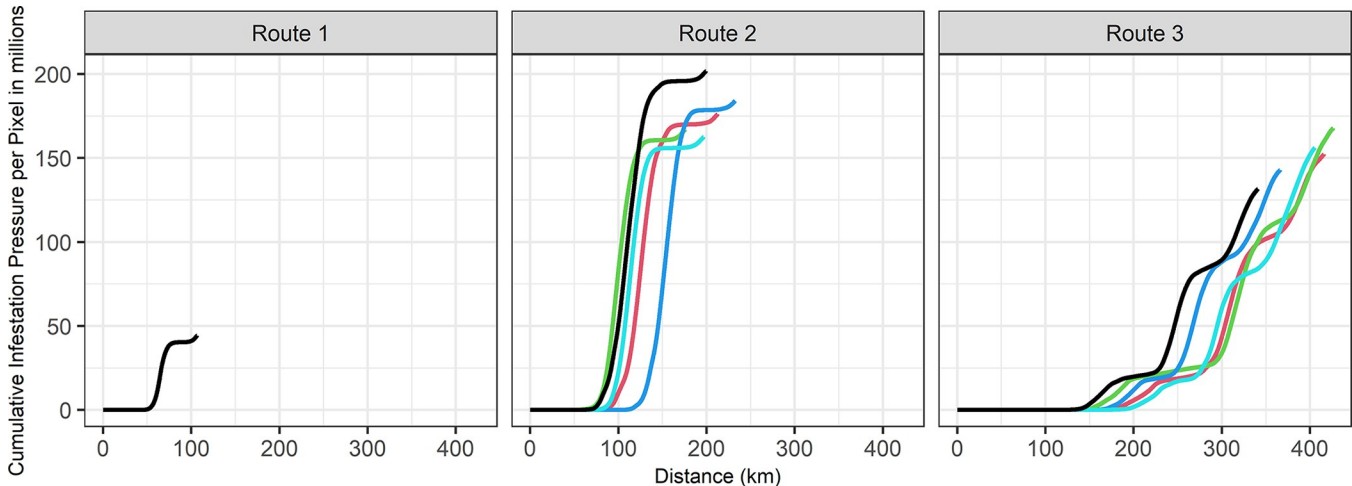

**Fig 5. Pixel-wise infestation pressure during juvenile sockeye salmon migration.** This figure illustrates the cumulative infestation pressures encountered in five representative simulations across three different migration routes. The five simulations in each route are signified by a distinct colour. Route 1 simulations display uniformity, reflecting deterministic migration patterns, while Routes 2 and 3 exhibit variations due to stochastic influences.

the mean abundance was 0.78, 2.01, and 3.09 for Routes 1, 2, and 3, respectively. Conversely, in the proportional relation, the mean abundance was 1.31, 5.02, and 4.53 for Routes 1, 2, and 3, respectively. The primary factor contributing to sea lice infestation on fish in Route 2 was

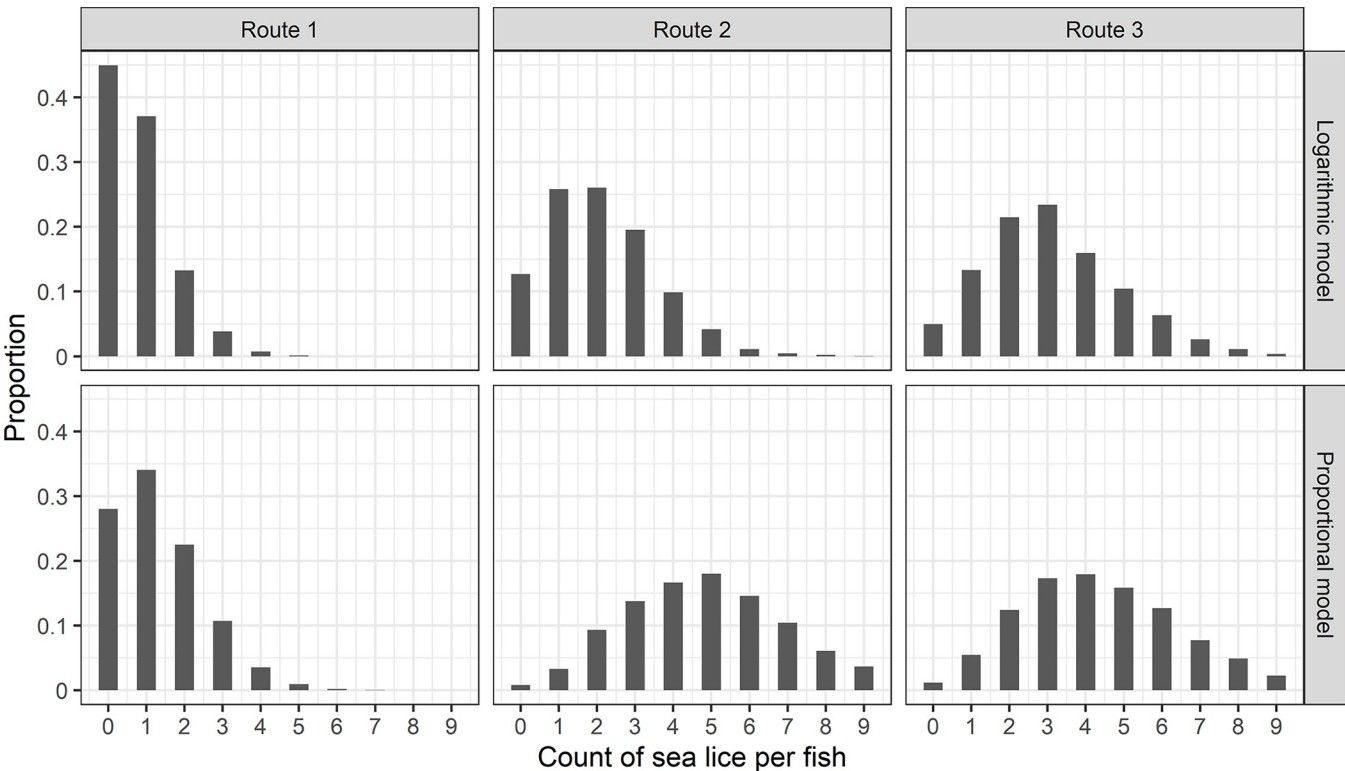

**Fig 6. Sea lice abundance variation across three migration routes.** Histograms represent the distribution of sea lice infestations for juvenile sockeye salmon across three different migration routes, based on 3000 simulations. The upper panels were generated using the logarithmic model, whereas the lower panels were produced using the proportional model.

the heightened total infestation pressure due to its closer proximity to salmon farms. In contrast, the elements enhancing sea lice infestation on fish through Route 3 were its longer migration distance and a slower progression rate, which extended the duration of exposure to lice from farms. In the logarithmic model, although Route 2 experienced greater total infestation pressure compared to Route 3, the markedly extended migration period of Route 3 resulted in a higher average lice abundance than Route 2. Conversely, in the proportional model, the impact of total infestation pressure was more pronounced than the duration of exposure, leading to a higher lice abundance in Route 2 than in Route 3. Additionally, the average lice abundance across all three routes was higher in the proportional model than in the logarithmic model.

Upon further analysis of the modelling results from the logarithmic model, it was evident that stochasticity played a significant role in influencing the sea lice count per fish within each route. For Routes 2 and 3, the correlation coefficients linking the sea lice count per fish to total infestation pressure were 0.14 and 0.12, respectively. Similarly, when associating lice abundance with migration distance, the coefficients were 0.08 for Route 2 and 0.07 for Route 3. These modest positive values suggest that while total infestation pressure and migration distance had an influence on the sea lice count per fish, their roles were not paramount. Instead, stochastic encounters largely dictated infestation dynamics. The substantial variation in the sea lice counts per fish across all routes further underscores the inherent stochasticity of these infestation events (Fig 6).

### 3.3. Sensitivity analysis

The tornado graphs display the strength and direction of the relationship of each parameter to the model output, which was generated through the logarithmic model (Fig 7-1) and through the proportional model (Fig 7-2). In Fig 7A-1, bars representing number of sea lice, bandwidth, and Routes 2 and 3 extend to the right, showing that as the parameter values increase, the model output does as well. Conversely, bars representing swimming speed and progression rate extending to the left suggest an inverse linear relationship. Except for progression rate, the four parameters demonstrated a comparable influence on the model output regarding sea lice abundance.

In our sensitivity analysis for each route, the influence of individual parameters varied. In the analysis of Route 1 (Fig 7B-1), bandwidth stands out as the variable with the most significant impact on the model's output, distinguishing it from Routes 2 and 3. In Routes 2 and 3 (Fig 7C-1 and 7D-1), swimming speed is identified as the most impactful variable on the model, with progression rate being the least. Notably, in Route 2, number of sea lice was more influential than bandwidth, whereas in Route 3, bandwidth was more influential than number of sea lice. Another finding between Routes 2 and 3 is that swimming speed and progression rate had a more marked influence in Route 2 compared to Route 3. Conversely, the effects of number of sea lice and bandwidth were somewhat subdued in Route 2 relative to Route 3, even though the disparity was modest.

Unlike the sensitivity analysis with the logarithmic model, using the proportional model revealed a remarkable influence from the number of sea lice released from salmon farms, with a marginal impact of bandwidth (Fig 7A-2). As the proportional model assumed a fixed slope between infestation pressure and lice abundance, the number of sea lice generated a substantial influence on determining the infestation rate. Consequently, lice abundance was predominantly driven by the number of sea lice, showing minimal dependence on the spatial distribution of larval lice in this model framework. Thus, the impact of bandwidth was minimal in the proportional model.

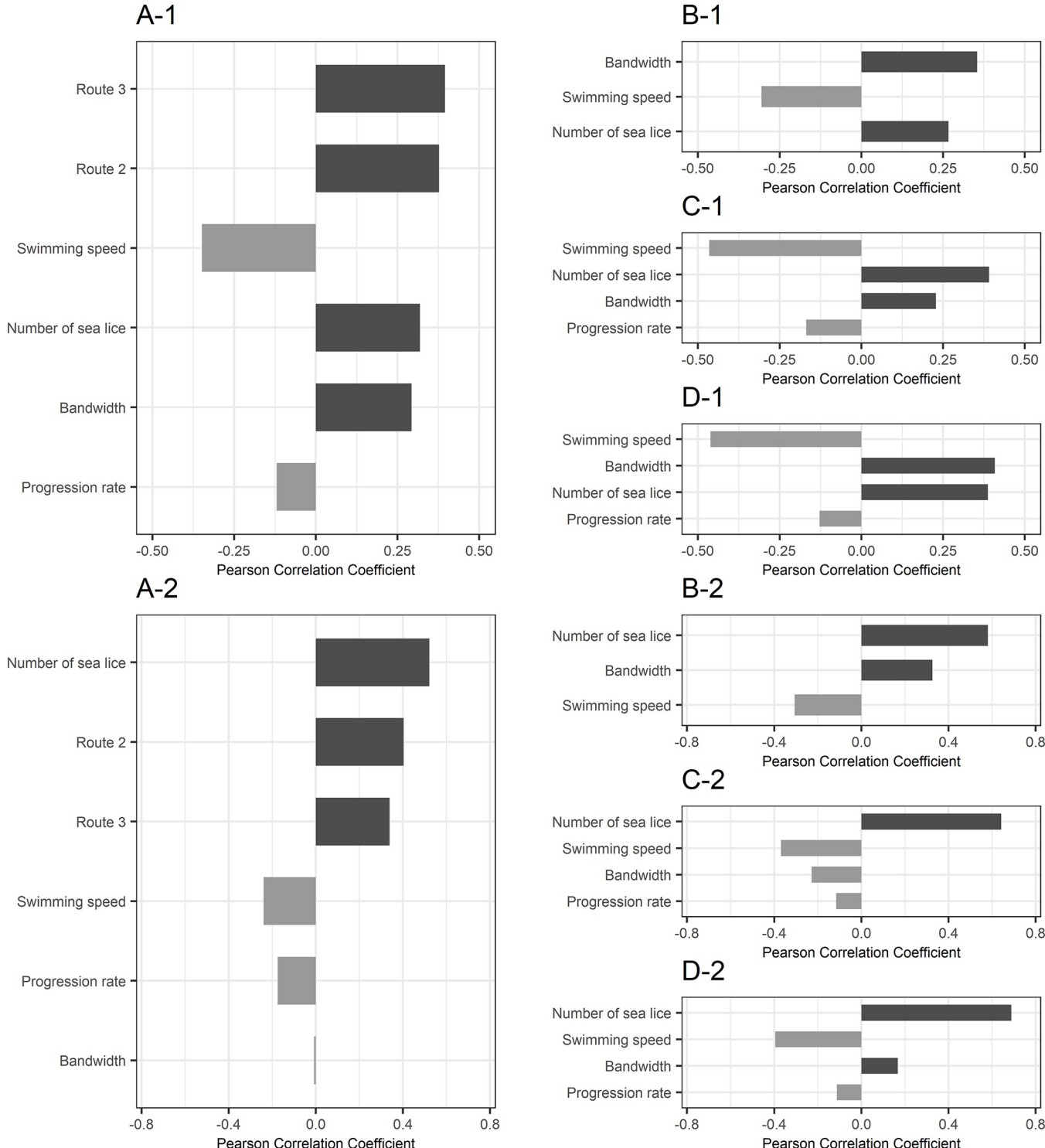

**Fig 7. Sensitivity analysis of parameters affecting salmon infestation using Pearson correlation coefficients.** (A) Comparative impact assessment of various parameters and route selection; Route 2 and Route 3 denote the relative impact of selecting these routes as opposed to Route 1. (B-D) Detailed impacts of parameters on infestation rates for individual routes: (B) depicts Route 1 where the progression rate is constant; thus, it is excluded from analysis, (C) and (D) show the variable influences on Routes 2 and 3, respectively. The numbers 1 and 2 in the headings correspond to the results from the logarithmic model and the proportional model, respectively. Bars oriented to the right signify a direct correlation between parameter value and model output. Conversely, bars to the left represent an inverse correlation. The length of the bar indicates the strength of the correlation, with parameters organized in descending order of influence.

## 4. Discussion

Building on the observations of juvenile sockeye salmon migration in the Discovery Islands, BC, Canada by [19], we examined sea lice infestations along different out-migration routes, quantifying their exposure to salmon farms. Our investigation focused on the impact of route-specific exposures on sea lice abundance in wild salmon. For this purpose, we utilized an ABM to simulate *C. clemensi* infestation during the out-migration of juvenile sockeye salmon in the region. The results from our simulations underscored that the sea lice abundance on wild salmon can vary markedly based on their specific migration routes, even within one region. This disparity in sea lice abundance, stemming from differing exposure levels to salmon farms, is mainly influenced not only by the number of sea lice originating from farms but also by the duration of exposure between the salmon and sea lice. Integrating insights from deterministic model of sea lice distribution and stochastic model of juvenile salmon out-migration enables to predict the potential risk of pathogen-host interactions and to estimate the probable lice loads on out-migrating juvenile salmon with a focus on spatial scale [20]. However, our findings indicate that inherent stochastic factors in lice infestation markedly influence infestation rates, leading to marked variability in sea lice counts across individual fish.

In our research, we refined the modelling of salmon migration patterns using progression rates, which accounted for the meandering pathways they often take, as opposed to previous studies that presumed direct trajectories [9, 14]. This adjustment provided a more realistic depiction, given that during their coastal migration, juvenile salmon frequently exhibit progression rates much lower than their actual swimming speeds due to varied behavioural strategies [8]. Notably, these rates varied depending on the migration routes [19]. For instance, salmon on direct routes like Route 1 progressed faster than those on more winding paths such as Route 3. This distinction is significant, as it suggests that salmon farms situated along meandering routes might present a higher risk of lice infestation compared to those on more linear paths.

The sensitivity analyses underscore the importance of swimming speed in determining the count of sea lice infested on fish. Previously, Samsing *et al.* investigated the effect of varying swimming speeds on infestation rates using cylindrical raceways [35]. They found that lice attachment was markedly influenced by swimming speed, with infestation levels being 2.5 times higher in moderate speed (17.0 cm/s) compared to high speed (36.7 cm/s) and 1.3 times higher compared to low speed (5.1 cm/s). It is important to consider that the swimming distances might not have been equal for the three speed levels. If an equal swimming distance was applied for all three groups of speeds, mean sea lice per fish would have been 4.6, 1.9 and 0.3 for low, moderate and high swimming speed, respectively. This supports our finding that slower speeds result in higher numbers of sea lice on fish. Moreover, the study observed that higher sea lice abundance at lower current velocities was due to an increased encounter rate and greater ease of attachment of larval lice to the host fish. This tendency for increased attachment at slower speeds, not considered in our model, could be another factor driving the higher sea lice abundance.

The results from sensitivity analysis discovered that swimming speed and progression rate had a greater influence on Route 2 than on Route 3. This indicates that parameters linked to the duration of salmon exposure to lice are more significant in shorter routes near farms. In contrast, the heightened impact of number of sea lice and bandwidth on Route 3, compared to Route 2, implies that factors related to infestation pressure are more impactful on longer routes located further from farms. This finding is attributed to the logarithmic relationship between infestation pressure and rate; variations in the lower infestation pressure of Route 3 are more noticeable than those in the higher infestation pressure of Route 2 [15].

The encounter rate function, denoted as Eq 3 and Eq 4, defines the relationship between the number of sea lice near fish and the likelihood of their encounter. Its significance in modelling results has been underscored in previous studies [9, 14, 15]. To develop this function, data has typically been drawn from two methods: observing lice infestation on migrating juvenile wild salmon through trawling [27] and using sentinel cages [14, 16]. Each method, however, presents its own set of advantages and limitations Notably, infestation rates from trawling data were found to be nearly ten times higher than those from sentinel cages [15]. In our study, we adopted the logarithmic model and proportional model, which were presented in Stige *et al.*, by using both trawling and sentinel cage data [15]. Moreover, the two distinct models, predicated on the presumed relationship between infestation pressure and lice abundance, generated differing modelling outcomes (Fig 6), highlighting the sensitivity of sea lice infestations on juvenile salmon to the underlying modelling assumptions. The critical nature of this relationship highlights the ongoing need to enhance the encounter rate function's accuracy. Improved precision in this function is essential for a deeper understanding of the dynamics of sea lice infestation.

As explained by the encounter-dilution effect [36], an individual fish's risk of parasitism increases with the rising number of parasites, but the increase is not linear. The encounter rate function, as delineated by Eq 3, shows a logarithmic escalation in lice per fish with increasing infestation pressure [15]. Thus, variations in lower infestation pressures have a more pronounced effect, while the impact lessens with changes in higher pressures. A possible reason for this phenomenon is that individual infective copepodids have specific attachment preferences for certain body parts of their host fish, resulting in heightened competition among sea lice for optimal attachment sites [37]. Moreover, a surge in attached lice might enhance the host fish's immune response, potentially curbing the likelihood of subsequent infestations [38]. Additionally, if we consider a hypothetical infestation zone surrounding each sea louse, the overlap of these zones becomes more likely as lice density increases. Based on the assumption that lice interfere with one another's ability to infest a fish when their zones overlap, this overlap reduces the likelihood of infestation in areas of high lice density [39]. This interference supports a theoretical basis for a logarithmic increase in the collective area of these zones with rising lice numbers.

To effectively address sea lice infestation on out-migrating juvenile salmon due to salmon farms, strategies could focus on either reducing lice release from farms or shortening the duration of exposure of wild salmon to these lice. Approaches to reduce number of lice may include regulating the number and biomass of farms and controlling sea lice levels on the farms. Meanwhile, reducing exposure time may involve managing the spatial density of salmon farms. Given the logarithmic relationship between infestation pressure and rate, farms in close proximity may result in a lower infestation rate on wild fish. However, spacing salmon farms apart is advantageous for reducing the transmission of sea lice between farms [40]. Further research is necessary to understand the collective impact of farm density. Additionally, the preferences of out-migrating fish for certain routes, influenced by factors like surface currents and wind, should be taken into account [41]. Sockeye salmon in the Discovery Islands, for example, show a preference for Route 2 over Route 3 [19]. These fine-scale migratory patterns could be crucial in efforts to minimize lice exposure from farms. Moreover, considering the geographical features when locating farms is significant; the features of juvenile salmon migration routes can affect their migration speed. Positioning farms along more direct coastlines, such as Route 1, where wild salmon tend to migrate more directly and potentially faster, could reduce the duration of lice exposure [19].

The development of acoustic telemetry as a novel tool for salmon tracking has facilitated detailed observations of salmon migration, not only within narrow rivers but also across

broader oceanic channels [42]. Our model of fish migration through the three routes was informed by the tracking observations of acoustic-tagged juvenile sockeye salmon by Rechisky *et al.*, which detailed their migration through the Discovery Islands at a fine scale, albeit with some degree of uncertainty [19]. Limitations in receiver array detection capabilities might lead to some tagged fish going undetected, thereby affecting the accuracy of migration data [19]. Additionally, the study makes assumptions about the fate of non-detected fish, suggesting potential causes such as handling stress or predation for their absence [19]. Nonetheless, the insights gleaned from this study furnish precise data on salmon migration at a fine scale through the application of advanced acoustic tagging, enriching future research and conservation strategies [19].

Our study examined the dynamics of *C. clemensi* infestation on sockeye salmon within the Discovery Islands, BC, Canada. Our study parameters were based on farm locations and wild salmon migration in 2017 and 2018. However, since 2020, the region has seen a systematic removal of open-net salmon farms due to a governmental prohibition on restocking [43]. Consequently, the aim of our research was not to evaluate the present-day risk of sea lice infestation on wild sockeye salmon in the Discovery Islands. Instead, our goal was to provide insights potentially transferable to similar ecological systems elsewhere. Furthermore, the parameter ranges assumed in the sensitivity analysis are crucial to the results obtained, as model outputs can vary significantly with different parameters [15]. To accurately assess the risk in a specific area, it is essential to have detailed data on the spatiotemporal variation in infestation pressure and a model specifically calibrated to observations of particular sea louse and salmon species. Therefore, the equations and parameter values related to the dynamics of sea lice infestation on wild salmon require further refinement to improve the accuracy of risk assessments for farm-origin sea lice.

In our research, we utilized an ABM to address the spatially stochastic patterns in salmon migration. Building on this, it would be advantageous to expand the application of ABM for modelling the spatial dispersal of larval lice [44]. This extension would involve integrating ABM with hydrodynamic modelling, moving beyond the sole use of a kernel density function for estimating the spatial distribution of larval lice numbers. Such a combined approach would enable to more accurately model the spatiotemporal variations in larval lice concentrations, adapting to the specific local environmental conditions [20].

## Acknowledgments

We are grateful to Dr. Derek Price for his thorough review of our manuscript and for offering invaluable feedback.

## Author Contributions

**Conceptualization:** Jaewoon Jeong, Gregor McEwan.

**Data curation:** Jaewoon Jeong.

**Formal analysis:** Jaewoon Jeong.

**Funding acquisition:** Jaewoon Jeong.

**Investigation:** Jaewoon Jeong.

**Methodology:** Jaewoon Jeong, Gregor McEwan.

**Project administration:** Jaewoon Jeong.

**Resources:** Jaewoon Jeong.

**Software:** Jaewoon Jeong.

**Validation:** Jaewoon Jeong.

**Visualization:** Jaewoon Jeong.

**Writing – original draft:** Jaewoon Jeong.

**Writing – review & editing:** Jaewoon Jeong, Gregor McEwan.

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
