## [Decision Letter · Decision Letter 0]

2 May 2024

PONE-D-24-06375Wild salmon migration routes influence sea lice infestations: An agent-based model predicting farm-related infestations on juvenile salmonPLOS ONE

Dear Dr. Jeong,

Thank you for submitting your manuscript to PLOS ONE. After careful consideration, we feel that it has merit but does not fully meet PLOS ONE’s publication criteria as it currently stands. Therefore, we invite you to submit a revised version of the manuscript that addresses the points raised during the review process.

We look forward to receiving your revised manuscript.

Kind regards,

Vanessa Carels

Staff Editor

PLOS ONE

Journal Requirements:

2. Thank you for stating the following financial disclosure: "Aquaculture, Biotechnology and Aquatic Animal Health Science Branch at the Fisheries and Oceans Canada provided a research supporting fund for this study. "

3. We note that Figures 1 and 3 in your submission contain [map/satellite] images which may be copyrighted. All PLOS content is published under the Creative Commons Attribution License (CC BY 4.0), which means that the manuscript, images, and Supporting Information files will be freely available online, and any third party is permitted to access, download, copy, distribute, and use these materials in any way, even commercially, with proper attribution. For these reasons, we cannot publish previously copyrighted maps or satellite images created using proprietary data, such as Google software (Google Maps, Street View, and Earth). For more information, see our copyright guidelines: http://journals.plos.org/plosone/s/licenses-and-copyright.

a. You may seek permission from the original copyright holder of Figures 1 and 3 to publish the content specifically under the CC BY 4.0 license.  

Reviewers' comments:

Reviewer's Responses to Questions

**Comments to the Author**

1. Is the manuscript technically sound, and do the data support the conclusions?

Reviewer #1: Partly

Reviewer #2: Yes

2. Has the statistical analysis been performed appropriately and rigorously? 

Reviewer #1: Yes

Reviewer #2: Yes

3. Have the authors made all data underlying the findings in their manuscript fully available?

Reviewer #1: Yes

Reviewer #2: Yes

4. Is the manuscript presented in an intelligible fashion and written in standard English?

Reviewer #1: No

Reviewer #2: Yes

5. Review Comments to the Author

Reviewer #1: Please see uploaded file.

Reviewer #2: Dear Authors,

Thank you for the submission of your manuscript for the publication consideration at PLOS ONE. The manuscript titled "Wild salmon migration routes influence sea lice infestations: An agent-based model predicting farm-related infestations on juvenile salmon" is drafted intelligibly and adhering to scientific principles. Although, the migration and infestation patterns are influenced by various parameters, the simplicity of the model and it's reliance on stochastic modelling eases the interpretation of the results and conclusions derived from it.

There are minor correction I would recommend to improve the manuscript.

1. Re-phrase the sentences

line 26-28 "Specifically, salmon traveling longer migration routes farther from farms may experience higher sea lice loads than those on shorter, farm-adjacent routes"

This sentence may mislead the readers given that the shorter route is already allocated the higher progression rate with zero alpha based on previous studies. The results may only discuss about the progression rate and infestation levels rather than their proximity to farms.

2. Abbreviations

line 58 ABMs -

Avoid plurals in abbreviations

3. Consistency

line 139 and 146

Fraser River and Fraser river

Adhere to one form when indicating the names of the places

4. Syntax

line 145

1st - superscript

line 198

"Details" - Please use the section number along with the section heading

line 339

"British Columbia"

Already defined as BC and yet it has been expanded again

line 552

"sockeye"

Capitalise the first letter of a sentence

I would recommend the authors do a thorough language and syntax checks prior to re-submitting this manuscript.

I am looking forward to read your amended manuscript.

Good luck

Reviwer

6. PLOS authors have the option to publish the peer review history of their article (what does this mean?). If published, this will include your full peer review and any attached files.

Reviewer #1: **Yes: **Leif Christian Stige

Reviewer #2: No

---

## [Author Response · Author response to Decision Letter 0]

17 May 2024

We appreciate the time and effort that you and the reviewers have dedicated to providing valuable feedback on our manuscript. We are grateful to the reviewers for their insightful comments on our paper. We have now been able to incorporate changes to reflect the suggestions provided by the reviewers, as indicated in the ‘bullet points’ below.

---

## [Decision Letter · Decision Letter 1]

3 Jul 2024

PONE-D-24-06375R1Wild salmon migration routes influence sea lice infestations: An agent-based model predicting farm-related infestations on juvenile salmonPLOS ONE

Dear Dr. Jeong,

Thank you for submitting your manuscript to PLOS ONE. After careful consideration, we feel that it has merit but does not fully meet PLOS ONE’s publication criteria as it currently stands. Therefore, we invite you to submit a revised version of the manuscript that addresses the points raised during the review process.

fix things pointed out by reviewer 1.

We look forward to receiving your revised manuscript.

Kind regards,

Arnar Palsson, Ph.D.

Academic Editor

PLOS ONE

Journal Requirements:

Additional Editor Comments:

Adhere to comments from reviewer 1

Reviewers' comments:

Reviewer's Responses to Questions

**Comments to the Author**

1. If the authors have adequately addressed your comments raised in a previous round of review and you feel that this manuscript is now acceptable for publication, you may indicate that here to bypass the “Comments to the Author” section, enter your conflict of interest statement in the “Confidential to Editor” section, and submit your "Accept" recommendation.

Reviewer #1: All comments have been addressed

2. Is the manuscript technically sound, and do the data support the conclusions?

Reviewer #1: Yes

3. Has the statistical analysis been performed appropriately and rigorously? 

Reviewer #1: Yes

4. Have the authors made all data underlying the findings in their manuscript fully available?

Reviewer #1: Yes

5. Is the manuscript presented in an intelligible fashion and written in standard English?

Reviewer #1: Yes

6. Review Comments to the Author

Reviewer #1: I find that the authors have addressed all my previous comments nicely.

I have a few additional comments after reading through the text anew.

1) I think the limitations of the infestation model should be clarified better:

- Around L 315 (“lice data”), it should be clarified that the infestation model is based on data on salmon lice Lepeophtheirus salmonis infestations on juvenile Atlantic salmon Salmo salar.

- Around L 625-630, where model limitations are discussed, I think it should be mentioned that to accurate assess the risk in a specific area, it would be necessary with detailed data on spatiotemporal variation in infestation pressure and an infestation model calibrated to observations of Caligus clemensi sea lice on juvenile sockeye salmon Oncorhyncus nerka.

This limitation is not relevant for the conclusions drawn in this paper, but relevant to mention since the model presents the model for the first time, and the purpose of the model as given on L 110-111 seems to open for applying the model for a somewhat wider purpose.

2) I find the text now generally very clear. The explanation on L 376-383 can however be made clearer by explaining that the random draw of values for the four parameters was done jointly, resulting in 3000 simulations where all four parameters varied randomly within the specified ranges.

3) Figure 7 legend needs to explain the difference between A-1 and A-2 etc.

4) L 584-588: An implicit assumption of this argumentation is that lice interfere with one another if their zones overlap, so that a louse has reduced chance of infesting a fish if it is close to another louse. Otherwise, intersecting zones do not reduce the expected number of infestations compared to non-intersecting zones (but the variance is increased). This assumption should be made explicit and the underlying mechanism explained, and if possible supported by references.

5) I noted a few possible minor errors:

L 181: this  these

L 252: “k and l”  “l and m”, for consistency with eq. 2?

L 293 – 295: “they” on L 295 does not point back on anything. Rewrite first sentence to start, e.g., with “Price et al demonstrated markedly...”?

L 322: Quotation marks around ‘proportional model’ are missing

L 411: This  These

L 428: the smaller  smaller

7. PLOS authors have the option to publish the peer review history of their article (what does this mean?). If published, this will include your full peer review and any attached files.

Reviewer #1: No

---

## [Author Response · Author response to Decision Letter 1]

22 Jul 2024

July 22, 2024

Journal: PLOS ONE

Manuscript ID: PONE-D-24-06375R1

Title: “Wild salmon migration routes influence sea lice infestations: An agent-based model predicting farm-related infestations on juvenile salmon”

Thank you for giving us the opportunity to re-submit a revised draft of our manuscript titled “Wild salmon migration routes influence sea lice infestations: An agent-based model predicting farm-related infestations on juvenile salmon” to PLOS ONE. We appreciate the time and effort that you and the reviewers have dedicated to providing valuable feedback on our manuscript. We are grateful to the reviewers for their insightful comments on our paper. 

1) I think the limitations of the infestation model should be clarified better:

- Around L 315 (“lice data”), it should be clarified that the infestation model is based on data on salmon lice Lepeophtheirus salmonis infestations on juvenile Atlantic salmon Salmo salar.

- We made clear the equation is based on Lepeophtheirus salmonis infestations on juvenile Atlantic salmon Salmo salar. 

- Around L 625-630, where model limitations are discussed, I think it should be mentioned that to accurate assess the risk in a specific area, it would be necessary with detailed data on spatiotemporal variation in infestation pressure and an infestation model calibrated to observations of Caligus clemensi sea lice on juvenile sockeye salmon Oncorhyncus nerka.

- We agree and updated the part to emphasize the need for detailed spatiotemporal data and a model calibrated to observations of specific sea lice and salmon species to accurately assess risks in specific areas.

This limitation is not relevant for the conclusions drawn in this paper, but relevant to mention since the model presents the model for the first time, and the purpose of the model as given on L 110-111 seems to open for applying the model for a somewhat wider purpose.

- With your comment, we clarified the applicability of our model for broader purposes in the part of Purpose. 

2) I find the text now generally very clear. The explanation on L 376-383 can however be made clearer by explaining that the random draw of values for the four parameters was done jointly, resulting in 3000 simulations where all four parameters varied randomly within the specified ranges.

- We revised the paragraph to clarify that 3000 simulations were conducted, each with random values for the four parameters.

3) Figure 7 legend needs to explain the difference between A-1 and A-2 etc.

- We updated the legend for Figure 7 to clarify the distinctions between labels 1 and 2. Additionally, we revised the corresponding sections of the text.

4) L 584-588: An implicit assumption of this argumentation is that lice interfere with one another if their zones overlap, so that a louse has reduced chance of infesting a fish if it is close to another louse. Otherwise, intersecting zones do not reduce the expected number of infestations compared to non-intersecting zones (but the variance is increased). This assumption should be made explicit and the underlying mechanism explained, and if possible supported by references.

- We revised the paragraph, explicitly mentioning the suggested assumption. 

5) I noted a few possible minor errors:

L 181: this  these

- We made the corrections as indicated. 

L 252: “k and l”  “l and m”, for consistency with eq. 2?

- We made the corrections as indicated. 

L 293 – 295: “they” on L 295 does not point back on anything. Rewrite first sentence to start, e.g., with “Price et al demonstrated markedly...”?

- We made clear who ‘they’ is indicating. 

L 322: Quotation marks around ‘proportional model’ are missing

- We made the corrections as indicated. 

L 411: This  These

- We made the corrections as indicated. 

L 428: the smaller  smaller

- We made the corrections as indicated.

---

## [Editor Report · Decision Letter 2]

8 Aug 2024

Wild salmon migration routes influence sea lice infestations: An agent-based model predicting farm-related infestations on juvenile salmon

PONE-D-24-06375R2

Dear Dr. Jeong,

We’re pleased to inform you that your manuscript has been judged scientifically suitable for publication and will be formally accepted for publication once it meets all outstanding technical requirements.

Kind regards,

Arnar Palsson, Ph.D.

Academic Editor

PLOS ONE
---

## [Editor Report · Acceptance letter]

12 Aug 2024

PONE-D-24-06375R2 

PLOS ONE

Dear Dr. Jeong, 

I'm pleased to inform you that your manuscript has been deemed suitable for publication in PLOS ONE. Congratulations! Your manuscript is now being handed over to our production team.

Kind regards, 

on behalf of

Dr. Arnar Palsson 

Academic Editor

PLOS ONE